# Sensopeptidomic Kinetic Approach Combined with Decision Trees and Random Forests to Study the Bitterness during Enzymatic Hydrolysis Kinetics of Micellar Caseins

**DOI:** 10.3390/foods10061312

**Published:** 2021-06-07

**Authors:** Dahlia Daher, Barbara Deracinois, Philippe Courcoux, Alain Baniel, Sylvie Chollet, Rénato Froidevaux, Christophe Flahaut

**Affiliations:** 1UMR Transfrontalière 1158 BioEcoAgro, Univ. Lille, INRAe, Univ. Liège, UPJV, JUNIA, Univ. Artois, Univ. Littoral Côte d’Opale, ICV—Institut Charles Viollette, 59000 Lille, France; d.daher@ingredia.com (D.D.); barbara.deracinois@univ-lille.fr (B.D.); sylvie.chollet@junia.com (S.C.); renato.froidevaux@univ-lille.fr (R.F.); 2Ingredia S.A. 51 Av. Lobbedez-CS 60946, CEDEX, 62033 Arras, France; a.baniel@ingredia.com; 3Oniris, StatSC, rue de la Géraudière, 44322 Nantes, France; philippe.courcoux@oniris-nantes.fr; 4INRA USC1381, 44322 Nantes, France

**Keywords:** bitterness, enzymatic hydrolysis, micellar caseins, off-flavours, peptidomics, random forests, regression trees, sensory analysis

## Abstract

Protein hydrolysates are, in general, mixtures of amino acids and small peptides able to supply the body with the constituent elements of proteins in a directly assimilable form. They are therefore characterised as products with high nutritional value. However, hydrolysed proteins display an unpleasant bitter taste and possible off-flavours which limit the field of their nutrition applications. The successful identification and characterisation of bitter protein hydrolysates and, more precisely, the peptides responsible for this unpleasant taste are essential for nutritional research. Due to the large number of peptides generated during hydrolysis, there is an urgent need to develop methods in order to rapidly characterise the bitterness of protein hydrolysates. In this article, two enzymatic hydrolysis kinetics of micellar milk caseins were performed for 9 h. For both kinetics, the optimal time to obtain a hydrolysate with appreciable organoleptic qualities is 5 h. Then, the influence of the presence or absence of peptides and their intensity over time compared to the different sensory characteristics of hydrolysates was studied using heat maps, random forests and regression trees. A total of 22 peptides formed during the enzymatic proteolysis of micellar caseins and influencing the bitterness the most were identified. These methods represent simple and efficient tools to identify the peptides susceptibly responsible for bitterness intensity and predict the main sensory feature of micellar casein enzymatic hydrolysates.

## 1. Introduction

The enzymatic hydrolysis of milk proteins displays a generally unpleasant bitter taste. The perception of bitter taste plays a crucial role in their use in various application fields. Indeed, the bitter flavour of extensively hydrolysed proteins has been and continues to be a major hindrance for their use. In addition, bitterness is sometimes combined with off-flavours that also appear during hydrolysis. However, these milk protein hydrolysates have significant advantages such as in sport nutrition where the use of these hydrolysates induces a very rapid release of amino acids in the blood, which may maximise muscle protein anabolism and facilitate recovery [1]. Moreover, these hydrolysates make it possible to boost muscle synthesis in sensitive subjects such as the elderly [2]. They are also used in clinical and infant nutrition where milk protein hydrolysates are recommended for a rapid supply of amino acids while ensuring low protein allergenicity. Indeed, the allergenicity of a protein is reduced or eliminated when the protein is hydrolysed into a low molecular weight peptide composition. Moreover, milk protein hydrolysates cater to the nutritional requirements of infants and toddlers, improving milk protein digestibility and reducing frequent spit-up.

For many years, scientists have performed important studies on explaining the appearance of bitterness in hydrolysates. For example, Murray and Baker were the first authors interested in the taste of protein enzymatic hydrolysates [3]. They found a bitter taste in enzymatic hydrolysates from caseins and lactalbumin, obtained with commercial proteinases, and a neutral taste in hydrolysates obtained from gelatine. Ichikawa et al. hydrolysed caseins, soy protein, ovalbumin and gluten with a proteinase from *Bacillus subtilis* and reported the development of a pronounced bitter taste with casein hydrolysates [4]. Various factors can influence the appearance of undesirable flavours in protein enzymatic hydrolysates, such as the nature of protein substrate(s) and enzyme(s), the hydrolysis duration, the selected pH and the temperature conditions. Concerning the casein proteins, β-, αS1- and κ-caseins produce the most bitter hydrolysates [5]. The causes for the bitterness were identified as early as 1970 by Fujimaki et al. and Matoba et al. [6,7] as being the presence of specific peptides rather than free amino acids in the protein hydrolysate. For example, the free forms of L-leucine and L-phenylalanine residues are bitter, with thresholds of 15–20 mM, but Leu-Leu or Ile-Leu and Leu-Phe are more than 10 times more bitter. Kim and Li-Chan (2006) and Iwaniak et al. (2018) confirmed that the bitter taste of peptides is determined by the presence of amino acids with high hydrophobicity [8,9]. According to Iwaniak’s data, the bitterness of peptides results from the presence of residues with bulky and branched side chains such as Leu, Ile, Val, Tyr, Phe and Trp. The bitterness of peptides also increases as the number of amino acids increases. Moreover, some structural characteristics, such as the diastereoisomer of the L series, the presence of a proline residue at the geometric centre and/or close to a basic amino acid, hydrophobic amino acids at N- and C-terminal positions in the peptide, and two and three residues of Leu, Tyr, Phe at the C-terminal of the peptide, influence the bitterness. In addition, it has been claimed that there are no bitter peptides for lengths greater than 25 residues [7].

In a previous study [10], the comparison between the sensory characteristics and the principal components of the principal component analysis (PCA) of mass spectrometry data reveals that peptidomics constitutes a convenient, valuable, fast and economic intermediate method to evaluate the bitterness of enzymatic hydrolysates as a trained sensory panel can conduct it. Nevertheless, to go further in the understanding of the peptide-related bitterness appearance/disappearance during the hydrolysis time, an enzymatic hydrolysis kinetic study gathering a sensory evaluation and a peptidomics approach combined with machine learning algorithms were carried out. Herein, we have studied the enzymatic hydrolysis kinetics of micellar caseins subjected to hydrolyses using commercially available and food-grade proteases, allowing the production of more or less bitter hydrolysates. Organoleptic characteristics, and more particularly the bitterness, were quantified for each sample collected during the kinetics using a trained sensory panel. Then, peptides generated during hydrolysis were characterised by a peptidomics approach combining the peptide chromatographic separation by reversed-phase high-pressure liquid chromatography (RP-HPLC), the detection and fragmentation of peptides by tandem mass spectrometry (MS/MS) and the mass data management. Finally, we studied the nature of the generated peptides and their influence in the appearance of bitterness during the hydrolysis process by using a method based on differential expression analysis, heat maps, regression trees and random forests. 

## 2. Material and Methods

### 2.1. Enzymatic Hydrolysis Kinetics

Micellar caseins (ratio micellar caseins/whey proteins (92:8)) were prepared by the Ingredia S.A. manufacturer (St-Pol-Sur-Ternoise, France) using industrial processes. These proteins were hydrolysed with the food grade enzymes (Table 1) Flavourzyme and Protamex that were obtained from Novozymes (Bagsvaerd, Denmark) and allowed the preparation of a kinetics named 109, and Promod 523MDP ™ and FlavorPro 937 ™ were obtained from Biocatalysts and allowed the preparation of a kinetics named 125 (Wales, UK).

The enzymatic hydrolyses were performed for nine hours using a confidential recipe. Overall, the protein solution of micellar caseins (92%) was diluted with distilled water to a concentration of 10% of total nitrogenous matter and brought to the desired pH by adding NaOH (4N). The necessary enzyme quantity was then added directly if it was in liquid form or solubilised in distilled water if it was in powder form. The hydrolysis monitoring was carried out by collecting data from pH, temperature and osmometry. Then, the degree of hydrolysis (DH) was determined using Nielsen et al.’s method based on the reaction of primary amino groups with ortho-phthaldialdehyde (OPA) [11], and the DH was calculated as previously described [10].

Samples were taken every hour and the enzymes were inactivated by heating at 98 °C for 3 min. About 1.5 L of hydrolysates was dried by atomisation using the Mini Spray Dryer B-290 from BUCHI (Rungis, France). The drying process was performed following the same procedure described previously [10]. Each hour, an aliquot of each hydrolysis was frozen at −20 °C before further analyses.

### 2.2. Analyses of Samples

#### 2.2.1. Sensory Analysis

Panel Composition and Training

The sensory analysis was carried out with a total of 19 healthy adults (12 females and 7 males, aged from 45 to 65 years old). They were enrolled in a training program, for 20 months, designed to identify and quantify the different descriptors chosen to characterise the hydrolysates. The descriptors were: (i) five odours (milk, fermented milk, rancid, soymilk and smelly), (ii) eleven flavours (bitter, sour, milk, sweet, mild, cheese, vanilla, salty, rancid, barn and whey) and (iii) five persistence flavours (bitter, sour, milk, sweet and cheese). The quantification of each descriptor was performed using a scale from 1 (low) to 7 (high). In this study, only bitterness data will be processed. Before starting this experiment, the performance of the assessors in terms of discrimination, repeatability and agreement was validated.

Tasting Conditions

The assessors evaluated the nine samples in a duplicate manner during four sessions (two sessions with four samples and two with five), for both kinetics. Those sessions were performed under standard sensory conditions (ISO 13299, 2003). Samples were presented in a sequential monadic way and their presentation order was based on a Williams’ Latin-square arrangement. Samples were dissolved in mineral water at a concentration of 10% of dry matter and 20 mL was presented in white plastic tumblers to each assessor and served at room temperature. The tests were performed in individual booths under white lighting and at 20 ± 2 °C. No time restriction was imposed on the assessors to perform this test.

Sensory Data Analyses

For both hydrolysis kinetics (109 and 125), sensory data were first assessed by a two-way ANOVA considering the samples and the consumers as factors and the bitterness scores as the dependent variable. A Duncan’s multiple range test (*p* ≤ 0.05) was performed to compare the samples two by two. These statistical analyses were computed using XLStat (XLStat 2020 1.1, Paris, France). 

#### 2.2.2. Mass Spectrometry: Sample Preparation and Peptide Characterisation Using HPLC-ESI-Qtof-MS/MS and Bioinformatics Treatment

The samples were prepared and analyzed in duplicate with the same method used in a previous study [10]. Briefly, peptides were purified and concentrated using a C18 solid phase extraction and 10 μL was separated using a reversed-phase high-performance liquid chromatography and an apolar gradient of 60 min: 1% ACN/0.1% of formic acid (FA) (*v/v*) for 3 min, then 1 to 30% ACN/0.1% FA (*v/v*) for 42 min, 30 to 95% ACN/0.1% FA (*v/v*) for 10 min and finally 95 to 99% ACN/0.1% FA (*v/v*) for 5 min. The analysis of eluted peptides was performed with a Synapt-G2-Si (Waters) mass spectrometer in sensitivity, positive and data-dependent analysis (DDA) modes (HPLC-MS/MS). Several quality control (QC) samples corresponding to (i) the mixture in equivalent volume of all C18-purified samples of both kinetics, (ii) the mixture of samples of kinetics 109 and (iii) those of kinetics 125 were also analysed at the beginning, middle and end of the HPLC-MS/MS analysis session.

Raw data from all HPLC-MS/MS runs were imported in Progenesis QI for proteomics software (Version 4.1, Nonlinear Dynamics, Newcastle upon Tyne, UK). First, data filtering was conducted before peak picking where a maximum charge of +4, a retention time defined between 5 and 50 min and a minimum intensity of 1000 were applied. Then, data alignment was automatically managed by Progenesis software using one of all QC runs as reference. Subsequently, manual alignment was performed if necessary, to optimise run alignment, and data normalisation was automatically performed for principal component analysis (PCA). The filtering criteria used for the statistical comparison of mass signals of HPLC-MS/MS runs were set as follows: (i) a maximum coefficient of ANOVA less or equal to 10^−10^ and (ii) only the identified peptides. Concomitantly, Progenesis software reported the quantitative evolution of peptides in terms of normalised abundance in the different hydrolysates. The variables used are derived from the comparison of peptide maps, i.e., the position of the isotopic massifs and their intensity. The reprocessing of mass spectrometry data and database searches to identify the peptides were performed via Peaks Studio version 10+ (Bioinformatics Solutions Inc., Waterloo, ON, Canada) using the UniProt database (10 September 2018) restricted to the complete proteome of *Bos taurus* organism. The parameters of mass tolerance thresholds, number of missing cleavage sites tolerated, choice of enzyme, and false discovery rate (FDR) were the same as previously [10].

### 2.3. Relationship between Sensory and Mass Data

The link between identified peptides and sensory perception of the samples was investigated using various methods: a heat map with differential expression analysis, and regression trees and random forest methodologies.

#### 2.3.1. Heat Map

A heat map was drawn from the MS-data corresponding to identified peptides from micellar caseins and their quantification in the 18 samples of both kinetics 109 and 125; the peptides corresponding to the features and the hydrolysis samples to the individuals. The heat map reflects the matrix data so that the values (normalised peptide abundance) are replaced by colour intensities ranging from yellow (low abundance) to red (high abundance). Cluster analysis was also carried out based on the heat map and the results were drawn as tree maps in the heat map [12]. 

Differential expression was also used to identify peptides that significantly influence the bitterness of hydrolysates. This latter was performed by merging kinetics 109 and 125. For the differential expression test, two groups were established: the group named “1” will be considered as more bitter and the “2” as less bitter. The split between these two groups was determined visually as follows: the samples were ranked in descending order of bitterness and the split was defined at the point where the greatest difference between two successive values was observed.

#### 2.3.2. Regression Trees and Random Forest

Regression trees (RTs) optimally subdivide the samples by a set of decision rules. These rules are constructed by iteratively separating the dataset with binary splits based on the choice of one predictor variable and an associated threshold value. The random forest (RF) algorithm generates multiple trees without pruning, improving the stability of the model. This is achieved by a double process of randomisation: (i) a random selection of the predictors at each node of each tree and (ii) each tree is grown on a different random data subset, selected by bootstrapping, i.e., sampling from the initial samples with replacement. The data portion used for the training phase is known as the “in-bag” data, whereas the rest is called the “out-of-bag” data. The latter will provide estimates of predicting errors [13]: the root mean square error of this predicted value is computed on the out-of-bag samples (RMSE_OOB_). 

In our case, RFs consist of modelling a sensory variable (bitterness descriptor) as a function of a number of predictors (presence or absence of peptides as well as their normalised abundance). The variables correspond to the 116 identified peptides. The aim here is to find out which peptides have a strong importance in understanding the intensity of the bitter descriptor. For this purpose, 50 forests of 5000 trees have been built. RFs will allow us to obtain the importance of each peptide and a confidence interval is computed around the importance of the peptides. All the peptides whose lower bounds of the confidence interval are greater than 0 are selected and are thus involved to predict the intensity of the studied attribute.

RTs and RFs have been carried out using language R 3.5.1 [14] and the R packages rpart [15] and random Forest [16].

## 3. Results

### 3.1. Influence of Hydrolysis Kinetics on the Sensory Characteristics of Hydrolysates

Figure 1 shows the evolution of bitterness intensity and DH during kinetics 109 (a) and 125 (b). ANOVA shows that samples are significantly discriminated (*p* ≤ 0.05) for both kinetics. 

Globally, the bitterness intensity (black line, Figure 1a) of the samples of the hydrolysis 109 decreases over the time. During the first three hours, the intensity is at its highest level and stagnates at the value of 5.20. Then, a decrease begins from 5.20 to 3.00 in 2 h of hydrolysis (between 3 and 5 h of hydrolysis) and remains stable at the bitterness level of 3 for the kinetics’ remaining time. After nine hours of hydrolysis, the DH (dotted orange line, Figure 1a) reaches a value of 50.8%. A consequent increase in the DH is observed between the 4th (13.9%) and 5th hour (44.1%), which is concomitant with the decrease in the bitterness intensity. A Pearson correlation coefficient (*r =* −0.933; *p ≤* 0.001) between DH and bitterness score suggests that the higher the DH, the less bitterness in the samples.

Concerning kinetics 125 (black line, Figure 1b), the bitterness appears to be stable over time. The minimum bitterness value is observed after two hours of hydrolysis with an intensity of 2.18 ± 0.37. A progressive increase in bitterness is observed after the 5th hour of hydrolysis and until the end of the hydrolysis, as indicated by the bitterness values which increase from 2.23 ± 0.46 to 3.59 ± 0.50. The DH (dotted orange line, Figure 1b) increases over time to reach the maximum value of 28.9% after nine hours of hydrolysis. Here, again a high increase, ranging from 10.9% to 22.6%, is observed between the 3rd and 5th hour of hydrolysis. However, contrary to kinetics 109, the bitterness intensity increase follows the DH increase, especially from the fifth hour of kinetics.

### 3.2. Peptide Characterisation and Peptide Abundance Evolution during the Hydrolysis

The HPLC-MS/MS raw data obtained for the 36 withdrawn samples (nine collected samples × two kinetics × two replicates) and the nine QCs (QC 109 × three replicates, QC 125 × three replicates and QC109–125 × three replicates) were imported in Progenesis QI for proteomics software. Among the 2635 peak picked mass signals, 479 mass signals have an ANOVA < 10^−10^, and among them, 116 mass signals were identified as milk protein peptides (Appendix A). These latter represent the global diversity, all hydrolysates combined, of identified peptides. Overall, 75 peptides from β-casein, 19 from α-S1 casein, 10 from α-S2 casein, 9 from kappa-casein, and 3 from β-lactoglobulin and no peptides from α-lactalbumin were identified. Between 114 and 116 peptides were identified per sample collected during the kinetics. The size features of identified peptides are: (i) a length comprising between 6 and 23 amino acids with a length mean of 11 amino acids and (ii) a molecular mass mean of 1303.37 ± 335.46 Da.

Peptides identified from β-casein corresponded mainly to three protein regions: Y75-G109, A116-F134 and T142-V224. The α-S1 casein- and α-S2 casein-peptides corresponded to three protein regions (G25-G48, L114-M138 and P192-P212) and two protein regions (L111-N130 and R185-A204), respectively. The κ-casein- and β-lactoglobulin-peptides corresponded to two protein regions (F39-G60 and F76-L95) and one protein region (V57-L73), respectively. 

PCA was performed using the 116 identified peptides (shown as light grey numbers in Figure 2) whose amino acid sequences are gathered in Appendix A. Figure 2 shows the first two principal components and illustrates the correlations between the 36 withdrawn samples. These principal components #1 and #2 explain 82.16% of the variance, and in such PCA, the more distant the groups, the more different in terms of peptide population. In the biplot presented in Figure 2, the technical replicates (same colour points) of each sample (including QCs) are close to each other, as can be seen with the examples shown with a red arrow on the PCA, indicating good technical repeatability. Moreover, the QCs of kinetics 109 (in yellow) are found at almost equal distance between the two groups formed by kinetics 109, and it is the same for those of kinetics 125 (in dark to light blue) and the QC of kinetics 109–125, which are found between the three groups represented on the PCA.

An agglomerative hierarchical clustering (AHC) allowed us to display three groups on the PCA: (i) a group circled in brown (top right) gathering samples 109-1, -2, -3, (ii) a group circled in burgundy red (top left) gathering samples 109-4, -5, -6, -7, -8, -9, and (iii) a central group circled in blue gathering all samples of kinetics 125. Notably, the evolution, according to the hydrolysis time, of peptide heterogeneity is clearly evidenced on the PCA of T1 to T9 of kinetics 125, which moves from right to left (from dark blue to light blue). As for the sensory analysis, a difference is observed between samples 109-1, -2, -3, which are significantly more bitter than the other samples of the kinetics. The samples of kinetics 125 are positioned between the two groups of kinetics 109 and thus appear to have peptide sequences common to both groups and with intermediate normalised abundances. 

The Progenesis QI software uses the peptide identities and their MS-based abundance data to generate an explicit picture of the evolution of normalised abundance of the peptide during hydrolysis kinetics (Figure 3a,b). As illustrated in Figure 3a,b, the peptides FVAPFPE (αS1-CN (39–45)) and LYQEPVLGPVRGPFPI (β-CN (207–222)) are more abundant during the first three hours of kinetics 109 (left part of curves), and conversely have negligible normalised abundance in the other samples collected during kinetics 109 and 125. On the other hand, Figure 3b shows normalised abundance curves according to hydrolysis times of peptides LQYLYQGPIVL (αS2-CN (111–121)) and YPFPGPIPNSLPQN (β-CN (75–88)), more abundant during kinetics 125. The latter are not or only very weakly present in samples 109-1, -2, -3, considered as the most bitter, suggesting that they do not bring significant bitterness to the samples. They would therefore not be responsible for the difference in bitterness between samples 109-1, -2, -3 and the others.

### 3.3. Relationship between Generated Peptides and Bitterness during Hydrolysis

#### 3.3.1. Heat Map

The heat map presented Figure 4 shows the differences in terms of normalised abundances between the samples of both kinetics in a more visual way than a table. On the heat map, the peptides are grouped in rows and the samples withdrawn during the hydrolysis kinetics (109 and 125) in columns. The peptides are divided into two groups: A’ and B’ (left dendrogram) and the samples are divided into two groups: A and B (top dendrogram). 

In order to identify the peptides responsible for the difference in bitterness, a differential expression analysis was performed by merging kinetics 109 and 125. Among the 18 samples, two different groups were formed according to the bitterness scores obtained by sensory analysis (group #1 and group #2). Group #1 includes the samples 109-1, 109-2, and 109-3, which all had a bitterness intensity greater than 3.91, and group #2 includes the remaining 15 samples of both kinetics with an intensity equal to or lower than 3.91. Among the 116 identified peptides, only 54 are significant (*p* ≤ 0.05—noted with an asterisk (*) in Figure 4) which means that they contribute to the difference in terms of bitterness between groups #1 and #2. 

The peptide group B’ corresponds to the 54 significant peptides discriminated by the differential analysis, and explains the difference in bitterness between the samples in groups A and B. The peptide group A’, corresponding to the non-significant peptides of the differential analysis, explains the difference between the samples of kinetics 125 and 109 (without the samples 109-1, 2 and 3). The sample group A corresponds to the three samples 109-1, 109-2 and 109-3 with the highest bitterness intensities according to the sensory data obtained by the panel. These three samples are well differentiated on the heat map: the red colour (corresponding to the peptide group B’) and the yellow colour (corresponding to the peptide group A’) rectangles on the left show that for the first samples of hydrolysis 109, we have a relatively high normalised abundance of group B’ peptides compared to group A’. According to their normalised abundance, these peptides as well as their quantity tend to explain the high bitterness of the first samples at the beginning of the hydrolysis. The second group of samples (B) includes the subgroup of the samples of kinetics 125 in the following order: 125-2, 125-1, 125-3, 125-4, 125-5, 125-8, 125-9, 125-6, and 125-7, and the subgroup of the remaining samples of kinetics 109: 109-4, 109-8, 109-9, 109-5, 109-6 and 109-7. The samples of the kinetics 109 subgroup with the exception of 109-1, 2 and 3 are characterised by high normalised abundances of the A’ peptide group. The kinetics 125 subgroup samples are characterised by high normalised abundances of the first eight peptides of group A’ and intermediate normalised abundances of the remaining peptides.

#### 3.3.2. Regression Trees and Random Forests 

RT and RF were also performed by merging the samples of kinetics 109 and 125. The importance of peptides as predictors of the main taste characteristic of enzymatic hydrolysates, namely bitterness, is presented in Figure 5. The measure of this importance quantifies the contribution of each peptide to the prediction of the sensory profile. Based on the confidence interval, we noticed that more than three quarters (94) of the peptides influenced bitterness. In order to build an accurate and parsimonious model and to identify the best subset of predictors from the 94 pre-selected predictor variables, we reduced the number of predictors by adding the peptides sequentially, from the most important to the least important one: all possible subsets from 5 to k variables are considered in turn. For each subset, an RF is constructed with the same parameters as before. The quality of the prediction was computed for each model generated and evaluated using the RMSE_OOB_ index. The lower it is, the better the quality of the prediction will be. The optimal RF was obtained for 22 peptides with a RMSE_OOB_ of 0.3405. The latter therefore corresponds to the first 22 peptides presented in Figure 5.

The model obtained with these selected 22 peptides was applied to both kinetics for predicting hydrolysate bitterness profiles (Figure 6). Figure 6 represents the evolution of the bitterness intensity of samples during kinetics 109 and 125 over time, with the observed values (full lines associated with full circles and triangles) and the predicted values (dotted lines associated with red empty circles and black triangles). The prediction is quite good, with a mean error of 0.34 for the predicted perception of bitterness and a correlation of 0.93 between observed and predicted values. 

Thanks to the 22 peptides, an optimal RT has been built (Figure 7). The first peptide which splits the initial 18 samples into 2 groups was FVAPFPE at a normalised abundance threshold value of 2,431,772. The three samples (node 7, *n* = 3) with a value higher than this threshold were considered as the most bitter. These latter are the three first samples of kinetics 109 (109-1, 109-2 and 109-3). On the other hand, fifteen hydrolysates with a FVAPFPE value below this threshold were grouped together (nodes 3, 5 and 6 with, respectively, 6, 8 and 1 sample(s)). Then, when the normalised abundance of NIPPLTQTPVVVPPF was lower than 56,200.67, six samples were separated from the others and revealed the least bitterness perception. A last split was performed another time with the peptide FVAPFPE and beyond the abundance of 656,107.8. The hydrolysate was again considered more bitter when the peptide abundance was higher than this value.

## 4. Discussion

The enzymatic hydrolysis of caseins is especially known for the appearance of bitterness, which is a hindrance to their use in the agri-food industry [17]. The first step of this study was to analyse the bitterness profile of two different hydrolysis kinetics called “kinetics 109” and “kinetics 125”. For these given hydrolysis conditions, the sensory evaluation, driven with a trained sensory panel, reveals that significantly different bitterness levels are obtained depending on the hydrolysis time. This latter is more visible for kinetics 109, for which we obtained a significant drop in bitterness after about three hours of hydrolysis, reaching an intensity of 3. It is well known that the development of a specific sensory profile in protein hydrolysates depends on the protein source, enzyme specificity and the conditions of the hydrolysis [18,19]. Based on all these rules, we selected and combined some specific enzymes known in the literature to aid in the development of low bitterness hydrolysates for applications in nutrition and the agri-food market. These enzymes are Flavorpro 937MDP ™, a mixture of endo- and exoproteases, which according to the manufacturer Biocatalysts, has been developed to remove the excessive bitter-tasting peptides produced when using animal and bacterial proteases; Promod 523MDP ™, an endoprotease with a bromelain activity, which is effective in the production of highly digestible proteins [20]; Protamex, which also substantially reduces bitterness when hydrolysing caseins, as indicated by the company Novozymes [21]; and finally Flavourzyme, which contains both endo- and exoprotease activities and has shown its efficiency in obtaining less bitter milk protein hydrolysates compared to other enzyme preparations [22]. This efficiency has been linked to the presence of high exopeptidase activity within this preparation [23,24]. Indeed, the exopeptidases cleave peptides from their C- or N-terminal extremities, allowing a reduction in the bitterness due to terminal hydrophobic amino acid residues, for example [25].

The sensory study of kinetics 109 and 125 allowed us to monitor on one hand the evolution of certain sensory features, and on the other hand the physico-chemical parameters, such as the DH. Concerning the latter, it emerged that it was not always correlated with the bitterness of the hydrolysates, as already confirmed by a lot of studies [26,27,28,29].

The second step was to identify the peptides generated during the time course of hydrolyses and concomitantly quantify their normalised abundance. These mass spectrometry data were then analysed by performing a heat map combined with a differential expression analysis of both kinetics combined. These statistical tests are very often used in the presence of big data such as OMICS-type data [30]. This heat map brings an overview of the peptide abundance evolution during hydrolysis kinetics and an image of the proximity of the samples. For kinetics 109, the lower peptide cluster is very abundant at the beginning of hydrolysis, then decreases over time while remaining stable during the last hours of the hydrolysis. Combined with the differential expression, 54 peptides have been identified as responsible for the bitterness difference. These latter are the most abundant peptides in the first three bitterest samples 109-1, 109-2 and 109-3, except LKKYKVPQLE, VYQHQKA and LSQSKVLPVPQgKA. The evolution of peptide abundance during kinetics 125 seems to be almost identical for all the samples constituting it, except for samples 125-1, -2, -3, which would explain the stability of its bitterness over time.

One of the main strengths of this study concerns the use of RT and RF methodologies to predict the bitterness of samples. It is important to note that single trees are easy to interpret and to establish the relationships inside the dataset. However, they are unstable, and small perturbations in the dataset can completely change their structure. Therefore, for the prediction of the sample taste, the use of the whole RF is more convenient. In fact, we obtained a very satisfactory quality index with a value of 0.34, meaning that the bitterness intensity can be predicted with a confidence interval of 0.34. Such a value is a good estimation in sensory evaluation. Moreover, the results showed that the RF highlighted the importance of peptides in explaining the bitterness of the 18 samples. As shown in Figure 5, the 22 most influential peptides selected to construct the RF are among the 54 peptides most involved in the differentiation of the two sample categories #1 and #2 derived from differential expression. The simultaneous presence of this set of peptides and their abundance are the cause of the difference in bitterness existing between the samples. An additional verification was made with the BIOPEP database. Available online: http://www.uwm.edu.pl/biochemia/index.php/en/biopep (accessed on 10 February 2021) [31] and the literature to determine if some of those 22 peptides were already reported as being bitter. In this database, all data about the taste of the peptides were obtained from sensory studies described in the literature. The peptides YQEPVLGPVRGPFP, YQEPVLGPVRGPFPIIV, APKHKEMPFPKYPVEPF, MAPKHKEMPFPKYPVEPF, AMAPKHKEMPFPKYPVEP and PVLGPVRGPFP had already been identified [17,32,33,34]. The studies of Karametsi et al. highlighted the following bitter peptides: GPVRGPFPIIV and YQEPVLGPVRGPFPI [33], and those of Toelstede et al. the sequence GPVRGPFP [35]. These results show that their primary structure is similar to peptides LLYQEPVLGPVRGPFPIIV, LYQEPVLGPVRGPFPI, LYQEPVLGPVRGPFP, LYQEPVLGPVRGPFPIIV and LLYQEPVLGPVRGPFP that we have identified in this study, suggesting that this protein region of the β-casein is conducive to the release of bitter peptides. Besides that, the FVAPFPE peptide is already known to display angiotensin converting enzyme inhibitory activity [36], but no information on its taste has been given. However, a peptide of close structure “FFVAPFPEVFGK”, referenced in the BIOPEP database, has been identified as bitter. Moreover, a correlation between the bitter taste of a peptide and bioactivity has been demonstrated [19]. The VEELKPTPEGDLEIL peptide was also characterised as bitter by Spellman et al. (2015). No information was found on the other peptide sequences. Thus, a large majority of peptides highlighted by RFs have already been identified in the literature. However, it is worth to note that it is the combination of the presence and/or the absence and the association of different peptides which is responsible for hydrolysate’s bitterness.

However, this prediction model has a major limitation since it can only be used with the same enzymes used in this study. The use of RFs makes it possible to take into account the entire complex mixture represented by a peptide hydrolysate. Indeed, the synergistic and antagonistic effects that may exist between peptides and their impact on bitterness are considered.

## 5. Conclusions

The impact of bitterness on food rejection has been studied extensively. Therefore, the development of protein hydrolysates with low levels of bitterness is an essential challenge for their incorporation in various foods. The hydrolysates formulated here may be used in the development of future food formulations such as peptide-fortified ready-to-drink infant formulae and low pH beverages such as fruit juices, for instance.

The data generated may be employed to inform the selection of a certain type of enzyme preparation and target the degree of hydrolysis values to generate hydrolysates with adequate sensory properties. A peptide hydrolysate is a complex mixture where interactions between peptides can complicate their study. However, random forests appear to be a useful tool for their analysis. Moreover, the use of RT and RF methodologies allows us, on one side, to highlight peptides involved in the explanation of the bitterness of samples, and on the other side to predict the bitterness profile of a micellar casein hydrolysate.

## Figures and Tables

**Figure 1 foods-10-01312-f001:**
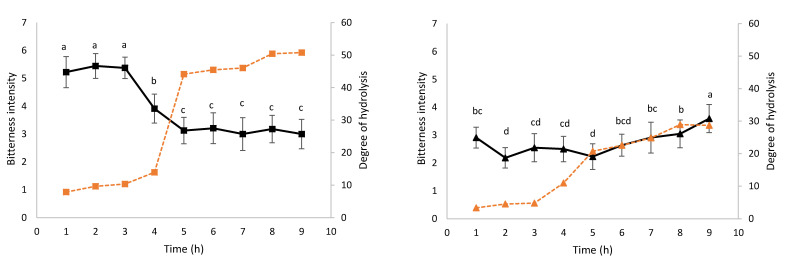
Evolution of bitterness and DH during hydrolysis kinetics 109 (**a**) and 125 (**b**). The black kinetics 109 (■) and kinetics 125 (▲), respectively. The dotted orange lines represent the evolution of DH for kinetics 109 (■) and kinetics 125 (▲), respectively. The bitterness intensity values from 1 (low) to 7 (high) are means +/− standard deviation (*n* = 2): the different letters (a, b, c) indicate means that significantly differ among the nine samples at *p* ≤ 0.05 according to Duncan’s multiple range test.

**Figure 2 foods-10-01312-f002:**
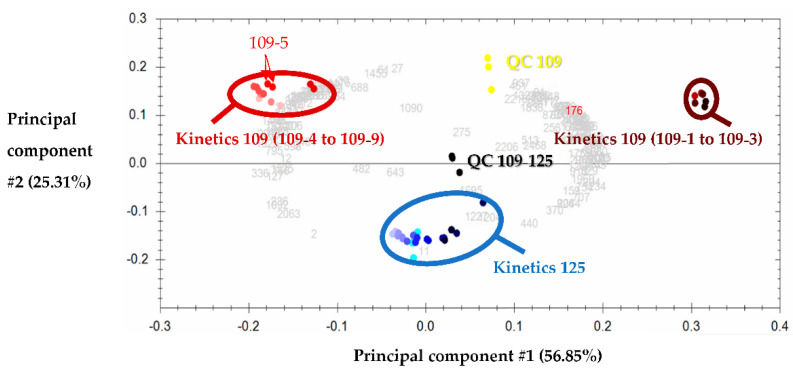
Principal component analysis corresponding to the 116 identified peptides of kinetics 109 and 125. In yellow the quality controls, corresponding to the equimolar mixture of the samples of kinetics 109 (QC 109), in blue “water green” those of kinetics 125 (QC 125) and in black those of all the samples combined (QC 109–125). Each QC appears as three replicates corresponding to an injection at the beginning, middle and end of the LC-MS/MS analysis. Each sample of both kinetics was analysed in replicates (samples with the same colour on the PCA as shown for the sample 109-5 (red arrows)), corresponding to a total of 36 samples: nine samples of hydrolysis kinetics 125 ranging from dark blue (125-9) to light blue (125-1) and nine samples of hydrolysis kinetics 109 ranging from brown (109-9) to very light red (109-1).

**Figure 3 foods-10-01312-f003:**
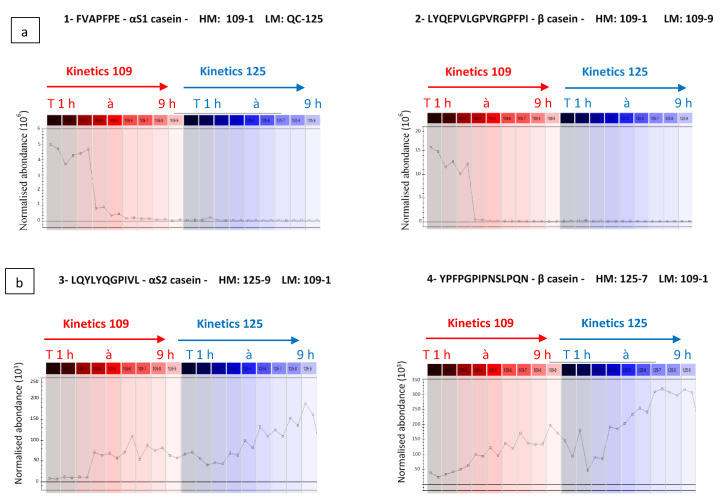
Examples of two opposite evolutions of the normalised abundance of peptides (**a**) whose abundance is highest at the beginning of kinetics 109 and (**b**) whose abundance is highest at the end of kinetics 125. The red and blue colour represent kinetics 109 and kinetics 125, respectively. (HM: highest mean; LM: lowest mean).

**Figure 4 foods-10-01312-f004:**
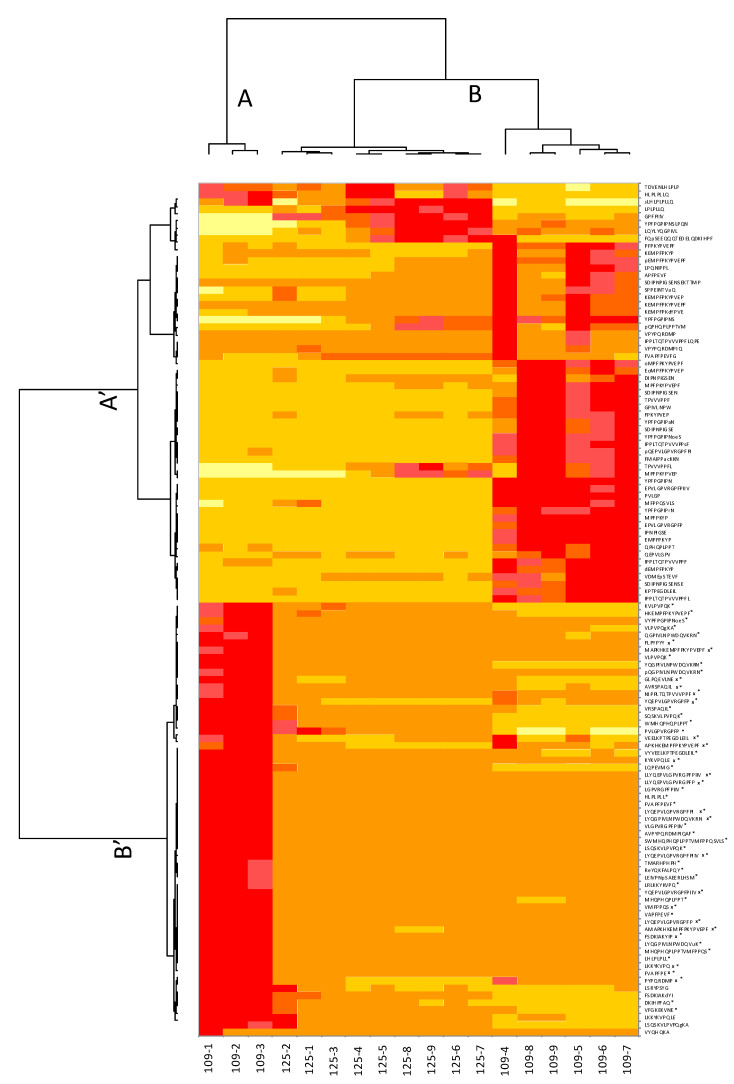
Heat map related to mass spectrometry data. The peptides marked with * correspond to the peptides that explain the significant difference between the two groups of hydrolysates obtained with the differential analysis (*p* ≤ 0.05). Peptides marked with an “x” are the peptides identified through random forests as the most influential in explaining the bitterness of hydrolysates.

**Figure 5 foods-10-01312-f005:**
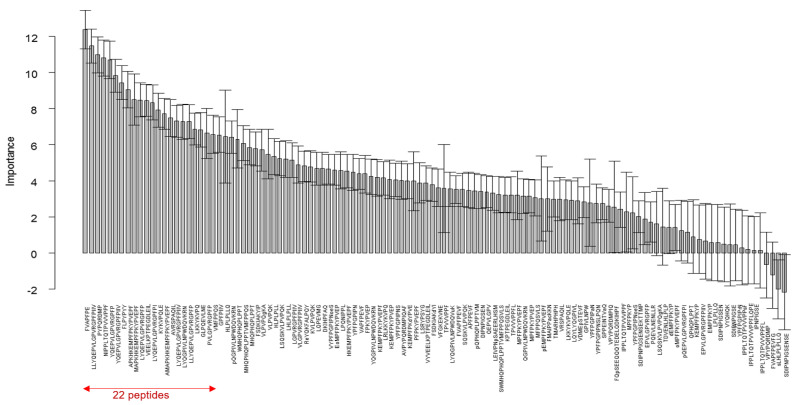
Random forests on the descriptor bitter: importance of the 116 peptides. The confidence intervals (95%) were obtained with 50 random forests of 5000 trees.

**Figure 6 foods-10-01312-f006:**
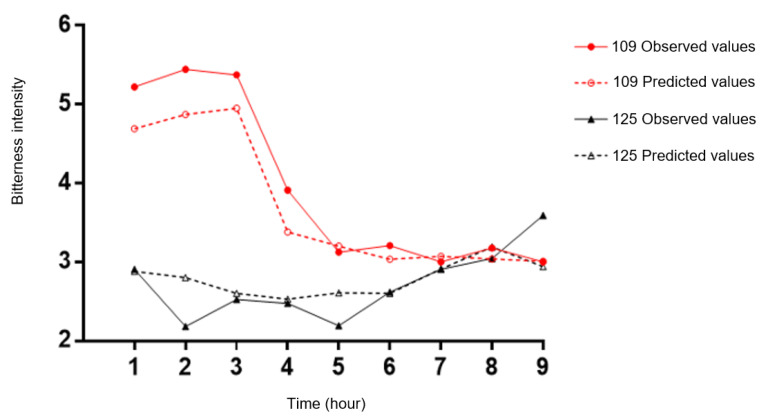
Prediction of the bitterness of out-of-bag (OOB) samples. The red colour represents the evolution of the intensity of samples during kinetics 109 and the black colour represents the samples of kinetics 125. The full lines with full-circles or -triangles represent the observed values and the open-circles and -triangles represent the predicted values.

**Figure 7 foods-10-01312-f007:**
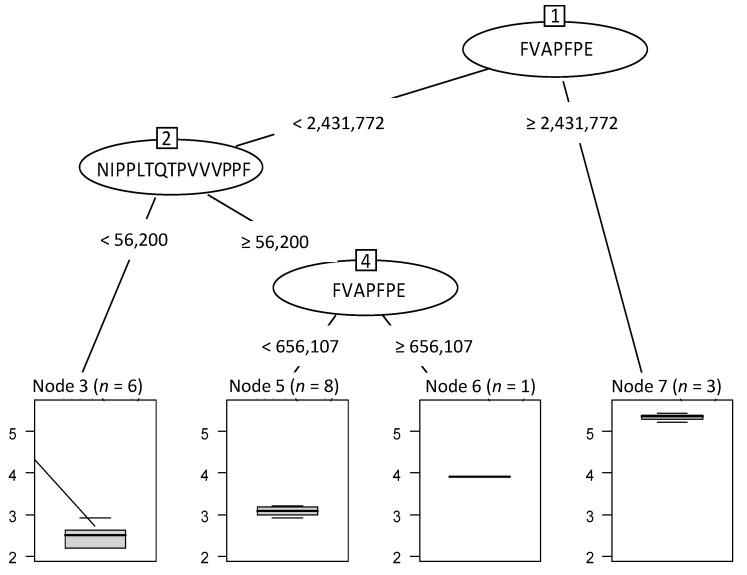
Optimal regression tree built to predict the hydrolysate bitterness from the 22 identified peptides. The boxplots represented below the tree show the intensity and the gradual evolution of bitterness. *n* = number of hydrolysates for each group defined by a different level of bitterness with sample reference number. Node 3 (*n* = 6) corresponds to hydrolysates 125-1, -2, -3, -4, -5 and -6; Node 5 (*n* = 8) corresponds to hydrolysates 125-7, -8, -9/109-5, -6, -7, -8 and -9; Node 6 (*n* = 1) corresponds to hydrolysate 109-4; Node 7 (*n* = 3) corresponds to hydrolysates 109-1, -2 and -3.

**Table 1 foods-10-01312-t001:** Characteristics of Novozymes proteases.

Proteases	Description	Activity *	Origin	Optimum pH	Optimum Temperature (°C)
Flavourzyme	exoprotease (aminopeptidase)/endoprotease complex	1100 LAPU/g	*Aspergillus oryzae*	5.5–7.5	50–55
Protamex	endoprotease (subtilisin)/serine endoprotease	1.5 AU-N/g	*Bacillus licheniformis* *Bacillus amyloliquefaciens*	7.0–8.0	50
Promod 523MDP ^TM^	endoprotease complex	1200 Bromelain GDU/g	*Ananas comosus*	5.0–7.0	45–55
Flavorpro 937MDP ^TM^	exoprotease (leucine aminopeptidase)/endoprotease complex	350 U/g	*Aspergillus oryzae*	5.0–7.0	50

* Leucine amino peptidase units per gram (LAPU/g); Anson unit per gram (AU-N/g); Gelatin digestion units per gram (GDU/g).

## Data Availability

Data is contained within the article or Appendix A.

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
