# Peer review of "Sensopeptidomic Kinetic Approach Combined with Decision Trees and Random Forests to Study the Bitterness during Enzymatic Hydrolysis Kinetics of Micellar Caseins"

_foods, 2021, doi:10.3390/foods10061312_

Round 1

Reviewer 1 Report

The manuscript “Sensopeptidomic kinetic approach combined to decision trees  and random forests to study the bitterness during enzymatic hydrolysis kinetics of micellar caseins” described an interesting investigation correlating the bitterness properties of micellar caseins hydrolysates to the amino acid structure of the generated peptides, also developing a prediction of correlation by using different statistical and bioinformatics tools. The manuscript is clear in both text organization and description of results and applied methodologies, for the latter often referring to authors’ very similar previous work, i.e. cited reference 11. This reference is not cited at all in the introduction section, where it should be conversely mentioned and described in its content, therefore better addressing the new results of the present investigation. Further comments and suggested alterations are following detailed:

-paragraph 2.1: two different kinetics were done, the 109 and 125, however it is not clear what was their difference, probably the enzyme recipe. The authors stated that the enzymatic hydrolyses were performed using a confidential recipe, however, out of the four enzymes described in text and Table 1, it should be specified how many of them were used in each kinetic and therefore if the kinetics difference is based on different mixtures of the all four enzymes or on their partial mixture. This limits the applicability of the present methodology in further studies and limits, as stated by the authors themselves, the applicability of the developed prediction model for protein hydrolysates bitterness, that is strongly dependent on the digestion procedure and the enzymes used. The authors should therefore better address the applicability of the present results in further studies, since different items have to be still investigated on the topic, as, i.e., correlating peptide secondary structures and bitterness as well as their interaction in concurring the final effect on protein hydrolysate taste.

-Supplementary Table S1: the number of decimals in monoisotopic molecular mass and in mass increment values of PTMs have to be reported according to the resolution power of the mass spectrometer used, therefore they need to be modified accordingly. Presently, the molecular mass and PTMs delta mass values show, differently, 7 and 2 decimals, respectively. Moreover, since describing the experimental results, Table S1 should be added of the experimental molecular mass values in addition to the monoisotopic theoretical ones for each peptide.   

Author Response

Answers to the Reviewer’s comments:

Dear reviewer,

Thank you for your useful comments and suggestions that undeniably improve our manuscript. For clarity, you will find below the answers and the list of changes made in accordance to your comments. All modifications are highlighted in yellow in the corrected manuscript.

Comments of reviewer #1

Open Review

English language and style

( ) Extensive editing of English language and style required
( ) Moderate English changes required
(x) English language and style are fine/minor spell check required
( ) I don't feel qualified to judge about the English language and style

Yes

Can be improved

Must be improved

Not applicable

Does the introduction provide sufficient background and include all relevant references?

( )

(x)

( )

( )

Is the research design appropriate?

(x)

( )

( )

( )

Are the methods adequately described?

(x)

( )

( )

( )

Are the results clearly presented?

( )

(x)

( )

( )

Are the conclusions supported by the results?

( )

(x)

( )

( )

Comments and Suggestions for Authors

The manuscript “Sensopeptidomic kinetic approach combined to decision trees  and random forests to study the bitterness during enzymatic hydrolysis kinetics of micellar caseins” described an interesting investigation correlating the bitterness properties of micellar caseins hydrolysates to the amino acid structure of the generated peptides, also developing a prediction of correlation by using different statistical and bioinformatics tools. The manuscript is clear in both text organization and description of results and applied methodologies, for the latter often referring to authors’ very similar previous work, i.e. cited reference 11. This reference is not cited at all in the introduction section, where it should be conversely mentioned and described in its content, therefore better addressing the new results of the present investigation. Further comments and suggested alterations are following detailed:

-paragraph 2.1: two different kinetics were done, the 109 and 125, however it is not clear what was their difference, probably the enzyme recipe. The authors stated that the enzymatic hydrolyses were performed using a confidential recipe, however, out of the four enzymes described in text and Table 1, it should be specified how many of them were used in each kinetic and therefore if the kinetics difference is based on different mixtures of the all four enzymes or on their partial mixture. This limits the applicability of the present methodology in further studies and limits, as stated by the authors themselves, the applicability of the developed prediction model for protein hydrolysates bitterness, that is strongly dependent on the digestion procedure and the enzymes used. The authors should therefore better address the applicability of the present results in further studies, since different items have to be still investigated on the topic, as, i.e., correlating peptide secondary structures and bitterness as well as their interaction in concurring the final effect on protein hydrolysate taste.

First question: We agree with your comment. Our previous article (Daher, D. et al., Foods 2020, 9, 1354, doi:10.3390/foods9101354) has then been added in the introduction (from line 74) as well as the main conclusions.

Second question: Effectively, you right. The difference between kinetics 109 and 125 relies on the recipe: the kinetics 109 was carried out using the commercial enzyme solutions Protamex and Flavourzyme whereas the kinetics 125 was performed with the commercial solutions Flavorpro 937 and Promod 523. These information have been added from line 93. For confidentiality reasons, the company Ingredia SA. does not wish to disclose the exact hydrolysis conditions (i.e. the amount of enzymes used…). The results obtained show us that the simple methods used to highlight the peptides, as well as their interaction, responsible for a various bitterness level, are useful in this field. The data generated can first help other scientists to select a certain type of enzyme preparations and target degree of hydrolysis values to generate hydrolysates with adequate sensory properties. The random forest obtained can then be used to predict the bitterness of future hydrolysates made with the 4 enzyme solutions used for the two kinetics.

-Supplementary Table S1: the number of decimals in monoisotopic molecular mass and in mass increment values of PTMs have to be reported according to the resolution power of the mass spectrometer used, therefore they need to be modified accordingly. Presently, the molecular mass and PTMs delta mass values show, differently, 7 and 2 decimals, respectively. Moreover, since describing the experimental results, Table S1 should be added of the experimental molecular mass values in addition to the monoisotopic theoretical ones for each peptide.   

We agree with your comment and the correction is done. We also added the m/z of each peptide.

Reviewer 2 Report

The paper “Sensopeptidomic kinetic approach combined to decision trees and random forests to study the bitterness during enzymatic hydrolysis kinetics of micellar caseins” contributes to the growth of literature for nutritionists as well as food producers offering products with enzyme preparations with adequate sensory properties.

The following items should be revised:

Abstract

Please add some describe the results and numerical data to the Abstract.

Line 71-84

Maybe better a clear definition of the aim.  This part of the text similar to the section on Methods.

Results

Figure 1  Value axis for "Degree of hydrolysis" need formatting without decimal places - similar to Axis values for "Bitterness intensity"

Figure 5  no figure

Figure 6 The font size is different than for Figure 1 axis captions and unclear

descriptions of the axis, e.g. for Figure 6 "Bitterness" and for Figure 1 "Bitterness intensity".  

The unit of time was not written. 

Figure 7  unclear data description    n=?  Node 3; 5 and 6  n=?

Author Response

Answers to the Reviewer’s comments:

Dear reviewer,

Thank you for your useful comments and suggestions that undeniably improve our manuscript. For clarity, you will find below the answers and the list of changes made in accordance to your comments. All modifications are highlighted in yellow in the corrected manuscript.

Comments of reviewer #2 :

Open Review

English language and style

( ) Extensive editing of English language and style required
( ) Moderate English changes required
( ) English language and style are fine/minor spell check required
(x) I don't feel qualified to judge about the English language and style

Yes

Can be improved

Must be improved

Not applicable

Does the introduction provide sufficient background and include all relevant references?

( )

(x)

( )

( )

Is the research design appropriate?

( )

(x)

( )

( )

Are the methods adequately described?

(x)

( )

( )

( )

Are the results clearly presented?

( )

( )

(x)

( )

Are the conclusions supported by the results?

( )

(x)

( )

( )

Comments and Suggestions for Authors

The paper “Sensopeptidomic kinetic approach combined to decision trees and random forests to study the bitterness during enzymatic hydrolysis kinetics of micellar caseins” contributes to the growth of literature for nutritionists as well as food producers offering products with enzyme preparations with adequate sensory properties.

The following items should be revised:

Abstract

Please add some describe the results and numerical data to the Abstract.

The correction is done.

Line 71-84

Maybe better a clear definition of the aim.  This part of the text similar to the section on Methods.

The paragraph starting now at line78 (due to the addition of further information) has been modified consequently for clarity.

Results

Figure 1  Value axis for "Degree of hydrolysis" need formatting without decimal places - similar to Axis values for "Bitterness intensity"

Figure 5  no figure

Thank you for this remark. Modifications have been made in 3.1.2. part for Figure 5 and Figure 6.

Figure 6 The font size is different than for Figure 1 axis captions and unclear

descriptions of the axis, e.g. for Figure 6 "Bitterness" and for Figure 1 "Bitterness intensity".  

The unit of time was not written. 

Thank you for your comments that, of course, we have taken into account.

Figure 7 unclear data description    n=?  Node 3 ; 5 and 6 n=?

We agree with your comment and we have amended the figure caption for clarity.

The "Nodes" define specific sensory profiles composed of one or several hydrolysates having the same specific sensory features (herein, the bitterness intensity). The number (n) of samples (as mentionned in the first version of the manuscript) is, in fact, the number (n) of hydrolysates

For example, the Node 3 gathers 6 hydrolysates (n=6) which are the hydrolysates  125-1, 125-2, 125-3, 125-4, 125-5 and 125-6.
